# OpenReview forum: "Evolving and Detecting Multi-Turn Deception using Geometric Signatures"
_ICLR.cc/2026/Conference — ICLR 2026 Conference Withdrawn Submission_

### Official Review · Reviewer_u9cX · 2025-10-30

**Soundness:** 3
**Presentation:** 2
**Contribution:** 2
**Rating:** 2
**Confidence:** 4

**Summary:**

The paper presents a unified pipeline to generate multi-turn deceptive questions and validates the dataset by a human study. The generation leverages multi-objective genetic prompt optimization with co-evolving mutation. The dataset can be used to help detect deceptive attempts in multi-turn conversations. Moreover, it identifies three geometric features, i.e., angular coverage, distance ratio, and linearity, as core indicators of deceptive attacks.

**Strengths:**

The paper contributes a new dataset to the community, including multi-turn deceptive questions, which can be valuable for training and evaluating models in this area. Although the dataset cannot fully cover all scenarios, I think the proposed framework enables it to be extended to other scenarios.

The paper presents a small set of signals that can capture the geometric properties of deceptive conversations, which is surprising and can inspire future research. This provides good explainability for the detection of deceptive questions.

**Weaknesses:**

There has been prior work on multi-turn deceptive question generation, or related topics, like multi-turn jailbreak generation. I am not sure why the paper choose to start from stretch and build their own dataset, rather than building on top of existing datasets. The paper should clarify this point.

The writing makes it a bit unclear, but I read this paper as a defense against existing systems. After all, the final product is a defense model that classifies deceptive questions. But the threat model is not clear in this case. Also, it does not really compare with alternative defense methods, like prompt rewriting, output filtering, etc. The paper should clarify the threat model and compare with alternative defense methods.

The paper also misses the discussion of efforts spent on Human-in-the-loop evaluation, which is important to evaluate the extendability of the dataset. I would appreciate it if the authors can present more details on how this part is done, e.g., expertise of the human evaluators, conflict of interests (authors, lab mates?), time spent, and cost.

The generated geometric features are interesting, but the paper does not provide a comprehensive evaluation of their robustness. Especially when the attack is adaptive, namely, optimizing prompts based on the defense, it is unclear how well these features can hold up. The paper should provide more evaluation on the robustness of these features.

**Questions:**

1. Can you clarify the threat model considered in this work? Is it purely a defense against existing systems, or is there a more specific scenario in mind?

2. How does the proposed defense model compare with alternative defense methods, such as prompt rewriting or output filtering? Have the authors conducted any experiments in this regard?

3. What was the expertise of the human evaluators, were there any potential conflicts of interest, and what was the time and cost involved in this evaluation? More generally, how scalable is the human-in-the-loop evaluation process for extending the dataset?

4. Have the authors evaluated the robustness of the identified geometric features against adaptive attacks? If so, what were the findings, and if not, do they have plans to conduct such evaluations in future work?

---

### Official Review · Reviewer_2YaF · 2025-10-30

**Soundness:** 3
**Presentation:** 3
**Contribution:** 3
**Rating:** 2
**Confidence:** 2

**Summary:**

This paper primarily addresses the issue that LLM safety systems struggle to detect "multi-turn, indirect" deceptive questioning. The authors propose a unified pipeline: first, they use a "multi-objective genetic prompt optimization" method to automatically "evolve" and generate a large set of realistic multi-turn deceptive question sets ; then, they demonstrate that this multi-turn deceptive intent leaves a "stable geometric footprint" in embedding space. Based on this hypothesis, they trained a lightweight classifier using only a few simple geometric features (such as angular coverage, distance ratio, and linearity). Experiments show this model is highly effective, achieving a high recall of 0.89, and remains robust even when questions are "reworded" or the conversation is shortened. This indicates that this advanced attack can be detected using only explainable geometric signals, without expensive end-to-end training.

**Strengths:**

Originality: The paper's originality lies in its unified pipeline, which uniquely combines a multi-objective evolutionary algorithm for generating novel multi-turn deceptive datasets with an innovative, lightweight detector based on a "geometric footprint" hypothesis. This approach of using simple, explainable geometric features to detect complex, coordinated multi-turn intent is a creative departure from common defenses that focus on single-turn heuristics, or end-to-end classifiers.

Quality: The work is of high quality, supported by a rigorous methodology. The authors did not just generate synthetic data; they validated it through a Human-in-the-Loop (HITL) study involving 52 participants, confirming that the data captures human-perceived deception. The detector itself was subjected to thorough robustness checks, including evaluations on reworded (lexical variation) and truncated (turn-constrained) datasets, demonstrating consistently high recall and strong generalization.

Clarity: The paper is written with high clarity. It clearly defines the specific threat model of multi-turn, indirect probing, using a concrete 5-question example about explosives to illustrate the concept . The novel geometric features (angular coverage, distance ratio, linearity) are explicitly defined and motivated . The experimental design is logical, cleanly separating the data generation/validation (Experiment 1) from the detector performance evaluation (Experiment 2).

Significance: This work is significant as it addresses a critical and nuanced vulnerability in current LLM safety defenses. Its primary contribution is a detector that is lightweight, explainable, and effective, making it highly suitable for practical, near real-time deployment. The finding that the detector maintains high recall (0.89) even when conversations are truncated to just three turns is particularly important, as it demonstrates a viable path toward early intervention before a harmful goal is achieved.

**Weaknesses:**

weakness 1: The Issue: The experimental design contains a significant confounding variable. All "deceptive" samples (60 sets) were generated around a single topic ("bomb") , while all "non-deceptive" samples (139 sets) were sourced from an entirely different dataset (Scale AI's general-purpose conversations).

weakness 2: The authors claim to test robustness to "lexical variation," but their method involved using Qwen-3B to generate paraphrases while enforcing an extremely strong constraint of "minimum 80% word overlap" This does not represent a genuine adversarial paraphrase. An 80% overlap implies the sentences are textually highly similar. A real attacker would use queries with far lower lexical overlap. Therefore, the reported performance on the "Reworded" set (F1 0.757) likely grossly overestimates the model's true robustness against a realistic adversary.

**Questions:**

Please refer to the weaknesses.

**Details Of Ethics Concerns:**

no Ethics Concerns

---

### Official Review · Reviewer_Qv3q · 2025-10-31

**Soundness:** 3
**Presentation:** 2
**Contribution:** 3
**Rating:** 2
**Confidence:** 3

**Summary:**

This paper presents a unified pipeline for generating and detecting multi-turn deceptive questioning of LLMs. The authors use multi-objective genetic algorithms to create synthetic datasets of indirect questions designed to elicit prohibited information, then develop a lightweight detector using geometric features in embedding space. The work addresses a timely and important problem in LLM safety.

**Strengths:**

**Novel and interesting approach**: Investigating multi-turn jailbreak defense from the perspective of geometric features in embedding space is creative and well-motivated.

**Preliminary promising results**: The initial experiments demonstrate potential effectiveness of the approach.

**Weaknesses:**

1.**Insufficient Model Generalizability** .The method's effectiveness is only explored on one target model. The authors should investigate the generalizability of their conclusions and methods across multiple models.

2.**Incomplete Discussion and Comparison with Existing Work and Baselines**.This paper proposes a synthetic data generation method to construct multi-turn jailbreak inputs. However, there already exist many multi-turn jailbreak studies, including works that decompose jailbreaks into multiple seemingly benign questions.  The authors should evaluate whether their detector can successfully defend against existing multi-turn jailbreak attacks.

Reference, for example: Chain of attack: a semantic-driven contextual multi-turn attacker for LLM

3.**Concerns About False Positive Rate on Non-deceptive Datasets** Although the paper reports test accuracy, considering the dataset imbalance (non-deceptive data comprises a small proportion), I worry this metric may not be sufficiently strong.

**Questions:**

Can the detector proposed in this paper defend against existing multi-turn jailbreak attacks?

---

### Official Review · Reviewer_sqqR · 2025-10-31

**Soundness:** 3
**Presentation:** 3
**Contribution:** 3
**Rating:** 6
**Confidence:** 3

**Summary:**

This paper addresses the underexplored problem of detecting multi-turn deceptive interactions targeting large language models (LLMs). Existing safety defenses are typically evaluated on single-turn prompts, overlooking the multi-turn strategies adversaries use to indirectly elicit prohibited information. The authors propose a unified framework that (1) generates realistic deceptive multi-turn question sets using multi-objective evolutionary optimization with co-evolving mutation operators and (2) detects deceptive intent through geometric signatures in embedding space. Three geometric features, angular coverage, distance ratio, and linearity, combined with pairwise similarity statistics are used to train a lightweight classifier that achieves high recall across several test conditions (base, reworded, and truncated scenarios). Human evaluation validates the realism of the generated data and demonstrates that deceptive multi-turn prompts exhibit consistent geometric patterns.

**Strengths:**

* Introduces a novel and timely perspective on multi-turn deception detection in LLM interactions.

* Develops a two-stage pipeline combining synthetic data generation and interpretable detection, both thoughtfully designed.

* The multi-objective genetic optimization with human-in-the-loop validation is well-motivated and produces plausible multi-turn deceptive examples.

**Weaknesses:**

* Rationale for geometric features needs deeper justification

The paper introduces angular coverage, distance ratio, and linearity as the core geometric features but does not sufficiently explain why these particular measures capture deceptive intent better than alternatives (e.g., manifold curvature, entropy of pairwise similarity, or graph centrality measures). The underlying intuition, that deception manifests as a geometric “footprint”, is intriguing but would benefit from theoretical or empirical support demonstrating these features’ discriminative value beyond heuristic reasoning. Including feature ablation studies or comparisons against alternative spatial statistics would make the claim more convincing.

* Limited generalization and dataset diversity

The experiments focus primarily on a single task, eliciting dangerous information about explosives, which may limit generalization to other domains of deception (e.g., social manipulation, misinformation, or medical advice). Although the approach is general, the lack of cross-domain validation makes it unclear how robust the geometric signatures are across topics and languages. The authors could improve the paper by demonstrating transferability of the detector to a different harmful intent category.

* Overemphasis on numeric performance without deeper interpretability

While quantitative results are strong (F1 up to 0.86 and recall up to 0.89), the discussion of why these metrics manifest is shallow. Visualization of the embedding geometry, for instance, through PCA or t-SNE plots showing deceptive versus benign clusters, could greatly enhance interpretability and substantiate claims of distinct geometric patterns.

* Limited theoretical grounding in cognitive or linguistic deception theory

The work would benefit from connecting the geometric patterns observed to linguistic or cognitive theories of deception, for instance, how deceptive multi-turn questioning reflects shifts in semantic coherence or topic exploration that manifest geometrically in embedding space. This would provide a richer conceptual explanation for the empirical results.

**Questions:**

Could the authors elaborate on the intuitive or mathematical rationale for selecting angular coverage, distance ratio, and linearity as geometric indicators of deception?

1. How do the results change if different embedding models (e.g., larger or domain-tuned sentence transformers) are used to compute geometric features?

2. How do the geometric features compare against simpler statistical or linguistic baselines (e.g., perplexity shifts, topic coherence measures, or dialogue entropy)?

3. Would visualizing the embedding trajectories of deceptive versus benign conversations help confirm that geometric separability exists?

---

### Note · Authors · 2025-11-13

I have read and agree with the venue's withdrawal policy on behalf of myself and my co-authors.